# Prevalence, Diversity and UV-Light Inducibility Potential of Prophages in *Bacillus subtilis* and Their Possible Roles in Host Properties

**DOI:** 10.3390/v14030483

**Published:** 2022-02-26

**Authors:** Haftom Baraki Abraha, Youbin Choi, Woobin Hyun, Jae-Won Lee, Hai Seong Kang, Min Seo So, Donghyun Shin, Jong-Hyun Jung, Desta Berhe Sbhatu, Kwang-Pyo Kim

**Affiliations:** 1Department of Food Science and Technology, Jeonbuk National University, Jeonju 54896, Korea; haftom@jbnu.ac.kr (H.B.A.); a01055414827@gmail.com (Y.C.); LEEEEL1010@gmail.com (J.-W.L.); sms2044@naver.com (M.S.S.); 2Indulgence Food Research Laboratory, Namyang Dairy Products Co., Ltd., Sejong 30055, Korea; mlnkop753@namyangi.com; 3Food Microbiology Division, Food Safety Evaluation Department, National Institute of Food and Drug Safety Evaluation, Cheongju 28159, Korea; rkdgotjd12@korea.kr; 4Department of Agricultural Convergence Technology, Jeonbuk National University, Jeonju 54896, Korea; sdh1214@gmail.com; 5Research Division for Biotechnology, Korea Atomic Energy Research Institute, Jeongeup 580-185, Korea; jungjh83@kaeri.re.kr; 6Department of Biological and Chemical Engineering, Mekelle Institute of Technology, Mekelle University, Mekelle P.O. Box 1632, Ethiopia; desta.sbhatu@mu.edu.et

**Keywords:** bacteriophages, *Bacillus subtilis*, prophages, prevalence, diversity, induction, in silico prophage analysis, insertion sites

## Abstract

*Bacillus subtilis* is an important bacterial species due to its various industrial, medicinal, and agricultural applications. Prophages are known to play vital roles in host properties. Nevertheless, studies on the prophages and temperate phages of *B. subtilis* are relatively limited. In the present study, an in silico analysis was carried out in sequenced *B. subtilis* strains to investigate their prevalence, diversity, insertion sites, and potential roles. In addition, the potential for UV induction and prevalence was investigated. The in silico prophage analysis of 164 genomes of *B. subtilis* strains revealed that 75.00% of them contained intact prophages that exist as integrated and/or plasmid forms. Comparative genomics revealed the rich diversity of the prophages distributed in 13 main clusters and four distinct singletons. The analysis of the putative prophage proteins indicated the involvement of prophages in encoding the proteins linked to the immunity, bacteriocin production, sporulation, arsenate, and arsenite resistance of the host, enhancing its adaptability to diverse environments. An induction study in 91 *B. subtilis* collections demonstrated that UV-light treatment was instrumental in producing infective phages in 18.68% of them, showing a wide range of host specificity. The high prevalence and inducibility potential of the prophages observed in this study implies that prophages may play vital roles in the *B. subtilis* host.

## 1. Introduction

All bacteria are prone to infection by the most abundant viruses on earth, which are called bacteriophages (also known as phages for short) [1]. Based on their lifestyles, phages are grouped into either virulent (phages that cause cell lysis after infection) or temperate (phages that cause lysogeny or cell lysis after infection) [2]. In the lysogenic lifestyle, the DNA of temperate phages is maintained as prophages within the host and is replicated as its chromosomal integrated form (integrated prophages) or in its circular or linear plasmid forms (plasmid prophages) to render the host as a lysogen [3,4]. The lytic genes of prophages are repressed, while some regulatory and lysogenic conversion genes continue to be expressed [5] to maintain the lysogenic state and to induce prophage-mediated host-phenotypic changes, such as super-infection exclusion, increasing pathogenicity, and expanding the metabolic capacities of the host [6,7].

About half of the sequenced bacteria harbor prophages [7] whose genome accounts from less than 1.00% up to 20.00%, depending on the type of the species [8]. The proportion of prophages in the genomes of *Bacillus thuringiensis* is 10.18% [9], while that of Group A *Streptococcus* (GAS) ranges from 7.10 to 12.40% [10]. The prophage of *Escherichia coli* O157:H7 strain Sakai accounts 16.00% [11], while that of *Streptococcus thermophilus* and *Streptococcus pyogenes* are 0.40 and 9.50%, respectively [12]. Moreover, the prophages of *Pectobacterium* spp. and *Dickeya* spp. represent less than 2.00% [13]. Not all prophages are able to induce infective phages because some may not be fully functional [14]. The literature shows that the prevalence of inducible functional prophages ranges from 0 (i.e., defective) to 42.00% [15,16].

*Bacillus subtilis* is a fast-growing, Gram-positive, spore-forming bacterium that is regarded as a model organism due to its importance in the study of the fundamentals of bacterial physiology and metabolism [17]. It is also considered as a key industrial organism due to its natural and engineered use in industrial production application, such as in food production (through fermentation), hydrolytic enzymes (*amylases*, *protases*), and fine chemicals (e.g., riboflavin) [18]. Despite the importance of *B. subtilis* strains in the food industry and biotechnology, the temperate phages SPbeta and phi3T are the two main phages that have been involved in research the most in past decades, and the *B. subtilis* 168 strain is the most frequently preferred host [19,20]. However, no data are available regarding the prevalence, diversity, and inducibility potential of the prophages in large collections of *B. subtilis* strains.

Comparative genomics studies can show the diversity of prophages. Studies into the prophage-encoding putative proteins and inducibility potential in *B. subtilis* can help us to understand their desirable and undesirable effect on their hosts as well as on their applications in the fermentation industry. The present study explored (*a*) the prevalence, diversity, characterization, and plausible roles of prophages in 164 sequenced *B. subtilis* strains via in silico analysis methods and (*b*) the UV-light inducibility potential of prophages in 91 *B. subtilis* collections and their prevalence. This work represents the first comprehensive study on the prevalence, diversity, and UV-light inducibility potential of prophages in a large collection of *B. subtilis* strains.

## 2. Materials and Methods

### 2.1. Collection of B. subtilis Genome Dataset and In Silico Prediction of Prophages

Complete genome sequences of 194 *B. subtilis* strains deposited in NCBI (https://www.ncbi.nlm.nih.gov/genome/browse#!/prokaryotes/665/) (accessed on 5 February 2021) were retrieved for putative prophage prediction. Following manual analysis, the strains of 30 genomes were dropped from further analyses for multiple reasons, such as prophage predictions that comprised >99.80% of the host’s genome (BS38 (NZ_CP017314.1), HJ0-6 (NZ_CP016894.1), and SG6 (NZ_CP009796.1)), the prediction of prophages with ‘N’ nucleotides (FB6-3 (CP032089.1)) and when strains shared identical strain names (with >98.00% symmetric identity) coupled with the prediction of identical/nearly identical prophages. Thus, the genomes of 164 strains (Appendix A) were considered for prophage prediction using two programs: PHASTER [21] and PhiSpy [22]. PHASTER is a web-based program that uses gene/protein hits in a given region to predict prophages as intact prophages (IPs), questionable prophages (QP), and incomplete prophages (InP) based on their completeness. It also annotates the prophages using Glimmer software for structural annotation and BLASTP for functional annotation [21], which were used for the analysis of the putative prophage-encoded proteins.

The integrated intact prophage regions predicted by PHASTER were then checked for whether they could be identified by the second prophage predicting program, PhiSpy, which operates on a Unix-based OSs platform. The PhiSpy program can identify prophages with and without sequence similarities to known phage sequences. The program was run with the default setting and using the *B. subtilis*_subsp._*subtilis*_str._168 training set distributed with it. The exact position of the end coordinates of the PhiSpy-predicted regions showed some degree of deviation compared to the PHASTER prediction due to variations in the prediction methods [23]; thus, the significance of their similarity was further checked using BLASTN. Only integrated IPs that were identified by both programs were included in the downstream analysis. Because PHASTER has the ability to evaluate and report the completeness of the prophages, the prophage genome sequences extracted using the end coordinates specified by PHASTER were considered as IP genome sequences.

### 2.2. Occurrence, Distribution and Prevalence of Prophages in B. subtilis

The occurrence of IPs in the 164 *B. subtilis* genomes and their distribution as either their integrated or plasmid forms was evaluated. All of the putative prophages were used to study their occurrence and distribution along with the genome sizes of the hosts. Only IPs were considered to estimate the prevalence, abundance (number of IPs per host genome), and proportion (genomic proportion of IPs) and to study the features of their genomes. The genomic proportion of IPs was calculated as “% = (Total IP Length ÷ Host Genome Length) × 100”.

### 2.3. Bioinformatic Analysis of B. subtilis Prophages

A comparative analysis of the IP genomes was carried out by generating average similarity matrices using Gegenees v3.1 with the 200/100 setting-fragmented BLASTN and TBLASTX comparison methods [24]. The phylogenomic relationships of the IPs were produced using the neighbor-joining method in SplitsTree4 [25]. The phylogenomic tree was then visualized using the iTOL (v5) tool (https://itol.embl.de/ (accessed on 10 November 2021)). A genome-scale dotplot was created using the Genome Pair Rapid Dotter (Gepard) [26].

### 2.4. Growth Conditions of Bacteria and Bacteriophages

*The B. subtilis* used for the prophage induction study included strains from the Korean Culture Center of Microorganisms (KCCM), Korean Agricultural Culture Collection (KACC), Korean Collection for Type Cultures (KCTC), American Type Culture Collection (ATCC) as well as lab isolates (Appendix A). Of the strains, the genomes of five strains were found in the NCBI database, namely JCM 1465 (NBRC 13719 (NZ_AP019714.1)), KCCM 32835 (NCIB 3610 (NZ_CP020102.1)), SRCM 102751 (NZ_CP028217.1), KCCM 11316 (ATCC 6633 (NZ_CP0349431.1)), and KCTC 2217 (168 (NZ_CP0110052.1)). *B. subtilis* were cultured aerobically in tryptic soy broth (TSB: BD, Sparks, MD, USA) or TSB agar (TSA: TSB enriched with 1.5% agar) at 37 °C. *Lactobacillus* strains were cultured in BD Difco™ *Lactobacilli* MRS media. During the course of the phage isolation and propagation procedures, TA soft agar containing nutrient broth (8 g/L), NaCl (86 mM), MgSO_4_.7H_2_O (0.8 mM), MnSO_4_ (0.3 mM), and CaCl_2_ (1.0 mM) (pH 6.0), SM buffer containing 50 mM Tris-HCl (pH 7.5), and 100 mM NaCl plus 10 mM MgSO_4_ were used as described in the works of Bandara et al. (2012) [27].

### 2.5. UV-Light Survival Rate of B. subtilis

To optimize a protocol for the prophage induction experiment, *B. subtilis* strains were exposed to a UVC light source (T-8C, 8W with 254 nm range, VILBER) at varying distances (i.e., 8, 9, and 10 cm) and exposure times (i.e., 1, 2, 3, 5, 10, 15, and 20 min) in the dark. After the addition of an equal volume of fresh TSB medium, cultures were kept in the dark for 2 h at 37 °C followed by dilution, plating, and incubation overnight at 37 °C for CFU counting. The survival rate was calculated using the plate count method presented in the reports of Djurdjevic-Milosevic et al. (2011) [28].

### 2.6. Prophage Induction up on Ultraviolet-Light Treatment of B. subtilis

*B. subtilis* strains were treated with UV light for the prophage induction experiments and for the supernatant preparations, and the strains served as indicator strains to test the infectivity of each supernatant by means of the dotting assay. An amount of 500 µL of exponentially growing *B. subtilis* culture was irradiated with UV-light, mixed with an equal volume of TSB, and incubated at 37 °C in dark for 2 h. Finally, the culture was centrifuged at 14,000× *g* for 5 min, followed by the collection and filter-sterilization (0.2 µm pore size) of the supernatants as outlined by Kim et al. (2012) [29]. All experiments were carried out in triplicate.

### 2.7. Infectivity, Prevalence of UV-Light Inducible Prophages and Phage Isolation

The infectivity and host spectrum of the induced prophages were determined by dotting the supernatants onto 91 indicator *B. subtilis* strains using the dotting assay method described by Kropinski et al. (2009) [30] or as indicated otherwise (Appendix A). An amount of 10 µL of supernatant was dotted onto pre-solidified TSB agar that had been overlayed with 300 µL of the indicator strain that had been grown overnight after being mixed with 4 mL TA soft agar. All supernatants a showing clear lysis zone (C) and some (n = 12) showing a turbid lysis zone (T) were considered for phage purification and evaluation. Moreover, the UV-light-treated supernatants that eventually produced infective phages were compared against their corresponding non-UV-light treated supernatants to check for any spontaneous prophage inductions. The non-UV-light-treated supernatants were prepared using the same method used for the preparation of UV-treated supernatants described in Section 2.5 above but without UV-light treatment.

### 2.8. B. subtilis Phage Host Range Analysis

Host range analysis was carried out against the *B. subtilis* strains and other purposely selected bacteria, including Gram-positive and Gram-negative as well as pathogenic and non-pathogenic species using the dotting assay method outlined by Ghosh et al. (2018) [31], in which 5 µL of phage samples were dotted on pre-solidified TA soft agar containing indicator strains. In addition to *B. subtilis*, five *B. licheniformis* (JCM 2505, SCC 125037, SDC 125016, SCC 123050, and SDC 125015), three *Lactobacillus* spp. (SRCM 100888, KACC 11451, and KACC 13877), five *B. cereus* groups (ATCC 1611, ATCC 14579, ATCC 13061, ATCC 21768, and ATCC 27348), one *B. thuringiensis* (ATCC 10792), one *B. mycoides* (ATCC 21929), one *Staphylococcus aureus* (ATCC 144458), one *Listeria monocytogenes* (Scott A), one *E. coli* (BW 25113), and one *Enterobacter sakazakii* (KCTC 2949) were included. Phages that showed identical host range lysis patterns were compared using the efficiency of plating (EOP) assay [32] and calculated as (PFU on Target Host Strain ÷ PFU on Propagation Host Strain) × 100.

### 2.9. B. subtilis Phage Large Scale Propagation Condition Obtimizations

Phage growth conditions such as media types (solid vs. liquid), temperature (30 °C vs. 37 °C), and multiplicity of infection (1 vs. 0.1) were optimized. Large-scale propagated phages were centrifuged at 10,000× *g* for 30 min and incubated overnight at 4 °C after the addition of 10% (*w*/*v*) polyethylene glycol (PEG 8000, Sigma, Saint Louis, MO, USA) and 0.5 M NaCl. They were then precipitated at 10,000× *g* for 30 min, the pellet re-suspended in SM buffer, filter-sterilized (0.2 µm pore size), and subjected to ultracentrifugation in six-density gradients of CsCl, vis-à-vis 1.7, 1.5, 1.45, 1.4, 1.3, and 1.2 g/mL of SM buffer at 166,900× *g* for 4 h using an ultracentrifuge (Soravel WX + ULTRA SERIES Centrifuge, Thermo Scientific, Waltham, MA, USA) at the Korea Food Research Institute (KFRI, Korea). Blue bands were withdrawn using a syringe from the centrifuge tubes (06750-AV tubes, Thermo Scientific, Asheville, NC, USA) and dialyzed against SM buffer by means of gentle shaking at 4 °C overnight followed by PFU checking and were stored at 4 °C [33].

### 2.10. Data Analysis

Data were analyzed using the ggplot2 package [32] provided in R programming [34] with its integrated development environment RStudio [35] (available at https://www.r-project.org/ and https://rstudio.com/ (accessed on 2 February 2022)).

## 3. Results

### 3.1. Prevalence and Distribution of Prophages in Sequenced B. subtilis Genomes

Prophages were predicted in 164 sequenced *B. subtilis* genomes and were used for prevalence and distribution analysis. The analyses showed that all of the strains harbored one or more types of intact prophages (IP), questionable prophages (QP), and incomplete prophages (InP). Whereas 92.07% (n = 151) of the strains contained IP and/or QP, 8% (n = 13) contained InP only. The analyses showed that 75.00% of the (n = 123) *B. subtilis* strains have IP, while 25.00% (n = 41) of them have no IP (Appendix A).

The sizes of the host genomes for most *B. subtilis* strains are confined from around 3.4 to 4.5 Mb and can contain all types of prophages. A relationship analysis between the type and abundance of putative prophages and host genome size showed no strong association. It is worth mentioning that the IPs were not found for strains with small-sized genomes (Figure 1).

When we look into the locations of the IPs on the replicons of the hosts, 93.50% (n = 115) of the strains have integrated IPs, 2.44% (n = 3) have plasmid IPs, and 4.07% (n = 5) have both integrated and plasmid IPs (Figure 2).

Initially, the PHASTER program was able to predict a total of 180 putative intact prophages (172 integrated and 8 plasmid forms) from 123 *B. subtilis* strains. In the genome of one strain (*B. subtilis* QB928), the PHASTER program identified two Ips with different scores from the same region, the reason for which is not clear. Thus, only one, the one with the highest score, was considered for further analysis. Overall, a total of 179 Ips (171 integrated and 8 plasmid forms) were predicted by PHASTER, and the details are provided in Appendix A.

Out of 171 integrated IPs predicted by PHASTER, 162 were re-identified by the second prophage prediction program, PhiSpy, which showed some degree of deviation in the exact locations of the start and end coordinates. Seven of the integrated IPs were not identified by PhiSpy and were hence excluded from the downstream analysis of the IPs. The remaining two integrated IPs predicted by PHASTER from the *B. subtilis* NCD-2 chromosome could not be checked by PhiSpy because its genome was not annotated. However, following manual inspection, most of the genes of these IPs were annotated to be phage proteins; thus, they were kept for further analysis. Taken together, the further downstream analysis was carried out by using 172 IPs, i.e., 164 integrated (162 verified by PhiSpy and 2 PhiSpy unverified) and 8 plasmid IPs (Appendix A).

### 3.2. Characterization of Intact Prophages in B. subtilis

Summary statistics were carried out for the genomic features of 172 IPs (Table 1). The mean genome length, genome proportion, protein number, and GC percent were found to be 55.02 kb, 1.86%, 71.33, and 41.63%, respectively. The attachment-site show-up and the number of *t*RNAs encoded by the IPs generated by the PHASTER program were evaluated. While the program identified the attachment sites for 56.39% (n = 97) of the IPs, 43.60% (n = 75) of the IPs did not show attachment sites. *t*RNA was only encoded in 8.14% (n = 14) of the IPs (Appendix A).

The abundance of IPs in each host genome was found to be 1 to 5, while their genomic proportion ranged from 0.01 to 7.18% compared to the genomes of their hosts. A total of 82 (66.66%) of the strains had one IP per genome (accounting <1.5% of host genome), while 27 (21.95%) of the strains had two IPs with genome proportions ranging from 1.5% to 4.5%. Nine (7.32%) strains harbored three IPs with a high genomic proportion ranging from 4.5 to 7.5% (Figure 3a). Generally, we observed that the genomic proportion of the prophages increased as the genome abundance per strain increased, as predicted.

The IPs were plotted against their length and GC-contents to analyze the features and associated genome distribution. (*a*) The IP genomes had genome lengths spanning from 20 to 150 kb with GC-contents ranging from 34 to 48%. (*b*) Nearly two-thirds of the IPs (n = 118) had genomes that were less than 50 kb long and broader GC-content distributions. (*c*) Some of the IPs (n = 28) had genomes that were 50 to 100 kb long with significantly broader GC-content distributions ranging from 34 to 48%. (*d*) Others (n = 26) IPs contain genomes that extended from 100 to 150 kb and that had a relatively narrow GC-content of 34 to 38% (Figure 3b).

The study into the relationship between the genome length of the IPs and their GC-contents revealed a slight negative association. When IPs have shorter genomes, they tend to have a higher GC-content and vice versa. The majority of the prophages with genome lengths that are less than 50 kb appeared to have a relatively higher GC-content (38 to 48%), while those with lengths greater than 100 kb tend to have lower GC-content (34 to 38%).

### 3.3. Diversity of B. subtilis 172 Intact Prophages in B. subtilis

Both nucleotide and translated amino acid sequence comparisons were carried out with the Ips at genome-scale using Gegenees software. A comparison heatmap revealed the presence of prophage clusters with identical/nearly identical and highly or less similarity scores. Following BLASTN comparison, the majority of the IPs (n = 153) could be assigned into 13 main clusters and were put into blocks. Identical or nearly identical and highly similar IPs (score > 70%) are depicted in green and t in black rectangles. Some IPs are similar (score > 40% and depicted in yellow and put in blue rectangles), while others are dissimilar (score <40% and depicted in red) (Figure 4a). Moreover, some of the IPs showed no similarity and existed as singletons.

Some members of the IPs of the cluster depicted in green (e.g., clusters H, I, and K) have similarity scores as high as >99%, implying their ubiquity, where their sequences are shared by numerous strains, rendering them to colonize a diversity of strains. Dissimilarity among the IPs might imply their distinctness across the strains. The details of the similarity scores used to build the heatmap and the members of each cluster are presented in Appendix A.

The TBLASTX-translated IP comparison analysis generated similar clusters to that of the nucleotide comparison method, but with reshuffling of few IPs from one cluster to another. Cluster labels that do not coincide with the BLASTN comparison in Figure 4a are shown by small letters in Figure 4b. Members of cluster ‘B’ in BLASTN comparison joined cluster ‘A’ in the TBLASTX comparison, while some members of cluster ‘A’ left their prior cluster and restructured into a new cluster (assigned as cluster ‘c’) in TBLASTX comparison mode (Figure 4b, Appendix A).

All 172 IPs sequences were concatenated into a single file in the same order as in the heatmap similarity matrix (Appendix A), with the exception of the singletons, which were concatenated to be at the end. Gepard with a sliding window of 10 nucleotides was used to generate a dotplot alignment, as shown in Figure 4c.

Looking into the annotations of the IPs, *Bacillus* phage SPbeta (NC_001884), *Bacillus* phage phi105 (NC_004167), *Brevibacillus* phage Jimmer2 (NC_041976), and *Brevibacillus* phage Jimmer1 (NC_029104) were among the topmost common phages that showed homology among the BLAST hits. The majority of the Ips that have the same most common phages hits were clustered together in the Gegenees analysis. To see the relationships at the genomic level, the genomes of the IPs were further compared based on the most common phages that they had in common as well as the *B. subtilis* temperate phage phi3T (KY030782.1) and the skin element defective prophage containing the *B. subtilis* 48 kb region found in NCBI (D32216.1), as seen in Appendix A.

### 3.4. Analysis of Insertion Sites of B. subtilis Intact Prophages

The insertion sites of the integrated IPs (n = 161) were analyzed to demonstrate the presence of common and distinct insertion sites as well as the 5′ and 3′ end genes of the prophages. Considering the position of both the 5′ and 3′ end coordinates, 138 of the IPs showed an intergenic site insertion, while five were intragenic. There were 18 other IPs that showed intragenic site insertion in their 5′ end coordinates only, and 10 others showed intragenic insertion at their 3′ end coordinates only. The DinB family- and sporulation-related proteins encoding the genes were among the frequently flanking genes of the prophage region on the left side. The spoIISC gene that encodes three-component toxin–antitoxin–antitoxin system antitoxin SpoIISC was the most frequent gene lanking the prophage regions from the right side. Most IPs of the same cluster tend to have the same or similar preferential insertion sites. Yet, different insertion sites were also observed in some IPs of the same cluster, suggesting their richness in diversity (Appendix A). Moreover, in the case of intergenic-inserted Ips, the 5′ end coordinate may be positioned at the 5′ end or downstream of the first prophage’s gene. Some prophages show intragenic insertions into RNAs, in which the prophages’ start and end coordinates are inserted within the tRNA sequences, including tRNA-Arg, tRNA-Val, tRNA-Thr, and rRNA-23S ribosomal RNA. Most IP members of a cluster prefer to have the same genes at their 5′ and 3′ ends (Table 2).

### 3.5. Phylogenomic Analysis of 172 B. subtilis Intact Prophages

Phylogenomic trees of Ips were constructed following the NJ method in SplitsTree4. The result showed that the Ips could be assigned into 14 clusters. Some of the clusters have many Ips while others have only a few. The phylogenomic tree analysis also showed the presence of singletons, indicating the diversity of the IPs (Figure 5).

### 3.6. Functional Annotation Analysis of Putative Proteins Encoded by B. subtilis Intact Prophages

Functional annotation analysis was carried out for 50 heterogeneous cluster-representative putative Ips that were selected based on their phylogenomic relationships. The analysis primarily focused on the gene products of the prophages in order to look into the plausible roles of the prophages in their host properties. The exercise demonstrated that the prophage-encoded putative proteins included various enzymes, such as proteases and peptidases, lipases, catalase, hydrolases, oxidases, and transferases. The prophages also encode putative sporulation-, bacteriocin-, immunity-, and arsenite and antibiotics resistance-related proteins. Some of the proteins were linked to transporters, while other products were associated with toxin–antitoxin and apoptotic control systems (Figure 6).

### 3.7. Optimization of UV-Light Treatment for Induction of B. subtilis Prophages

The effect of UV-light on the survival of the *B. subtilis* was optimized before the prophage induction study. UV-light exposure for 1 min at 8, 9, and 10 cm distances inactivated 65 to 80% of *B. subtilis*. The deactivation level increased from 75 to 87% upon increasing the exposure time by a factor of 1 (Figure 7). A linear relationship was observed between the survival rate and UV-light exposure distance for up to 2 min. As the exposure time increased to 2 min, the exposure distance tended to have less impact, suggesting the greater effect of the exposure time compared to the exposure distance on the survival of *B. subtilis*. Prophage induction increased as the viability of the host decreased [7]. It has also been documented that a survival rate of about 11% is suitable for meaningful growth comparisons [36]. In our case, since exposing the bacteria for 2 min at a 9 cm distance yielded an ~15% survival rate, that exposure setting (for 2 min from 9 cm) was adopted for prophage induction.

### 3.8. Examination of B. subtilis Supernatants for the Presence of Infective Prophages and Phage Isolation

Supernatants prepared from 91 UV-light-treated *B. subtilis* strains were dotted on 91 *B. subtilis* indicator strains using the spot assay method to assess the presence of induced infective prophages. The dotting assay profiles of all of the supernatants are presented in Appendix A. A given supernatant can display clear lysis zones in multiple or single indicators. Clear lysis-forming supernatants were considered for the next phage isolation steps. Some supernatants displayed clear zones in multi-indicators, providing an opportunity to consider alternative indicator hosts for phage isolation trials, given that clear lysis showing supernatants remains unresponsive to plaque formation tests or fails to produce sufficient phage titers. Infective phages were isolated from clear lysis-forming supernatants.

Some supernatants formed turbid lysis but not clear lysis. Ten (10) turbid lysis-forming supernatants and two (2) turbid and clear lysis-forming supernatants were considered for further phage isolation tests using indicators on which they showed turbid lysis to assess whether or not these supernatants are effective in producing functional phages. All of these supernatants failed to support plaque formation except the two (2) that form both turbid and clear lysis. Moreover, the UV-light-treated supernatants that formed the plaques were considered for comparisons against their corresponding supernatants that had been prepared without UV-light treatment. Some of the supernatants prepared from non-UV-light-treated *B. subtilis* strains were able to form a clear zone similar to the treated ones, suggesting the spontaneous induction of the prophages (data not shown).

### 3.9. Inducibility Potential of B. subtilis Prophages

Supernatants were prepared from 91 *B. subtilis* strains following UV-light treatment to assess the inducibility potential of the prophages. The spot assay test showed that 26.37% (n = 24) of the supernatants formed clear lysis, while 48.4% (n = 44) formed only turbid lysis zones. Likewise, the plaque-forming assay showed that 20.88% (n = 19) of the supernatants formed plaques. In conclusion, infective phages were isolated from 18.68% (n = 17) of the clear lysis-forming supernatants (Figure 8).

### 3.10. A Comparison of In Silico Predictions with UV-Light Induced Prophages

When searching for the *B. subtilis* strains to be used in both the in silico and induction studies, we were able to identify five strains (Table 3). The in silico prophage analysis showed that four of the strains contained at least one integrated IP, while one contained IPs in both their integrated and plasmid forms. The UV-light induction experiment showed that while all of the supernatants of the strains formed turbid lysis zones, only three display clear lysis zones tested on 91 indicator strains. During the plaque-forming assay, only one supernatant (NBRC 13719) showed plaques. The supernatants from the SRCM102751 and ATCC 6633 strains were able to form a clear lysis zone in multiple indicator strains but failed to form plaques during the plaque-forming assay.

### 3.11. Large-Scale Propagation Conditions and Host Range Analysis

Seventeen (17) *B. subtilis* phages were isolated and their respective host, and their propagation conditions are presented in the Appendix A. Most of the phages were propagated in large-scale liquid media at 37 °C. Few phages were propagated in solid media at 30 °C.

Host range analyses of the 17 phages were carried out in 50 indicators. Twenty-eight (28) indicators on which the phages showed clear lysis or no lysis were sorted for host range lysis pattern comparison (Table 4). Two groups of phages (green and blue) showed similar lysis patterns in all of the indicators, while the rest showed distinctly different lysis patterns. The member phages of the different groups appeared to have a narrower host range, while those that showed distinctly different lysis patterns tended to have broader host ranges. The phages with similar lysis patterns were further analyzed using relative efficiency of plating (EOP). However, they did not show satisfactory score variations to make any generalizations (data not shown).

Furthermore, the host range of the phages was investigated using 19 purposively selected non-*B. subtilis* bacterial species. These include Gram-positive bacteria, namely five *B. licheniformis* and *B. cereus*, three *Lactobacillus brevis*, and one *Staphylococcus aureus*, *Listeria monocytogenes*, *B. thurengenesi*, and *B. mycoides* and Gram-negative bacteria including one *Escherichia coli* and *Cronobacter sakazakii*. The phages showed no lysis on the non-*B. subtilis* strains, except for the *B. licheniformis* strains, on which some of the phages showed clear or turbid lysis (data not shown).

## 4. Discussion

The present in silico prophage study uncovered a high prevalence of IPs in a large collection of *B. subtilis* genomes. The study identified and examined 172 IPs from 123 sequenced *B. subtilis* strains using two prophage prediction programs, namely PHASTER and PhiSpy. Of the sequenced 164 *B. subtilis* strains included in this study, 75.00% of them contain IPs (Figure 8)—mostly in integrated form (93.50%). A total of 4.07% of the strains have both IP forms, while 2.44% only have prophages in their plasmid form, expanding the diversity of *B. subtilis* prophages. Previous studies documented that most temperate phages perpetuate their genomes through integration into the chromosomes of their hosts and that some maintain lysogeny via plasmid formation [37]. *Bacillus* phage vB_BtS-B83 *NC_048762.1) [38] and vB_BceS-IEBH [NC_011167.1] [39] are examples of experimentally proven plasmid prophages in *B. thuringiensis* host.

Prophages are particularly prominent in pathogenic bacteria [7,40]. Virulence factor-coding genes of pathogenic bacteria are often associated with prophages [41]. Previous in silico prophage prevalence and diversity analyses have mainly been focused on various pathogens. In one study using PHASTER, the prevalence of full-length or putatively full-length prophages in Gram-positive pathogenic *Pneumococcus* (n = 482) was 45.00%, and the total identified full-length prophages were 286 [42]. Studies using the same method reported prophage prevalence of up to 64.90% in Gram-negative *Pectobacteriaceae* phytopathogen (n = 57), with 37 intact prophages being identified [13], and up to 81.00% in Gram-positive *B. thuringiensis* (n = 61), in which 135 putative complete prophages were identified [9]. Likewise, studies using the multiplex PCR method showed 87.00% prophage prevalence in Gram-positive *S. aureus* pathogens [43] and 80.00% in Gram-negative *Flavobacterium psychrophilum* pathogens [44].

Relevant information on the prevalence of prophages in non-pathogenic bacterial species is limited. Thus, a higher or comparable prevalence of *B. subtilis* prophages with that of the aforementioned pathogens is interesting and of high value. Previously, it was suggested that lysogeny is frequent in fast growers, as they provide more resources for virion production [7]. Thus, it is possible that as fast-growing bacteria, in *B. subtilis* infection and lysognization across strains by temperate phages takes place frequently, resulting in a high prevalence of prophages.

High genetic diversity was noticed among the predicted IPs. Comparative genomics of the IPs revealed that the prophages demonstrated diverse but structured relationships, with most of them being grouped into 13 clusters and four distinct singletons. The presence of highly similar prophages in different *B. subtilis* strains could imply that such prophages may have high infective power, whereas the less similar ones and singletons may have lower infection ability. However, it was not easy to make generalizations because the number of hosts used for prophage identification or the number of related phages used for homology searches may not be comprehensive enough.

Detailed analysis of the gene annotations indicated by PHASTER showed that *Bacillus* phage SPbeta, *Bacillus* phage phi105, *Brevibacillus* phage Jimmer2, and *Brevibacillus* phage Jimmer 1 are the most common phages to 28 (16.28%), 25 (14.53%), 35 (20.35%), and 17 (9.88%) IPs. Upon the comparison of these IPs and phages, significant similarities were only able to be observed among some of the IP with the SPbeta and phi3T phages but not the others, supporting the high diversity of the IPs. Most of the Spbeta-related IPs were grouped into two clusters, L and M. Only about half of the phi105-related IPs could be assigned into cluster I, with the rest being distributed into multiple clusters. The observation that some of the Ips have the most common phages from different host species may indicate the limitation of the information provided by *B. subtilis* phages.

According to the current International Committee on Taxonomy of Viruses (ICTV) classification scheme, the *Bacillus* phage SPbeta and phi105 are the only two *B. subtilis* temperate phages classified to the genus *Spbetavirus* and *Spizizenvirus*, respectively, under the family *Siphoviridae*. Based on the ICTV criteria, to classify phages into the same genus, they should share >50% DNA sequence identity [45]. Our findings showed that only 30.81% (cluster L and M members) of the identified IPs shared significant sequence similarity with the *Bacillus* phage SPbeta and phi105, meaning that they may belong to the same respective genus. The majority of the other species showed no relatedness to either SPbeta or phi105, suggesting the rich diversity of the prophages in *B. subtilis* strains. It is important to note that the diversity of the IPs and their insertion site analysis results can vary with different prediction platforms. In fact, the two prophage prediction tools used in this study mostly showed variation in terms of the same IP end coordinates.

We used a large collection of *B. subtilis* strains (n = 91) to study the UV-light inducibility potential of *B. subtilis* prophages and 18.68% of them produced infective phages that can form clear plaques and be isolated. Hinting at the inducibility prevalence of the prophages in *B. subtilis*, five strains were studied in both in silico and inducibility studies. All of the strains appeared to have at least one IP in the in silico analysis, but following UV-light induction treatment, only one (20%) could produce plaques.

The inducibility potential of the prophages observed in this study is significantly lower than the prevalence of the IPs (75%), which are commonly identified from two prophage-predicting programs in the in silico analysis. During the induction study, some supernatants a showing clear lysis zone failed to form plaques and few plaques remained unresponsive in the subsequent phage isolation steps, indicating that some prophages might still be induced but somehow defective to further their lifestyles. A similar observation was reported in a study with *Staphylococcus suis*, where cells were lysed after the addition of mitomycin C but failed to form plaque in all of the tested indicators [41]. In addition, the gradual loss in the ability of the prophages to form plaque or phage particles and subsequently to lyse cells is documented [46]. This loss in ability occurs because of the inactivation of point mutations, genome rearrangements, modular exchanges, invasion by further mobile DNA elements, and massive DNA deletions [47]. Exploring the prevalence of inducible prophages in *B. subtilis* would require additional prophage inducibility studies using various inducing agents and a large collection of sequenced *B. subtilis* strains.

High IP prevalence in *B. subtilis* genomes and the inducibility potential traits observed in this study leads us to deduce that prophages may have considerable effects on their hosts. The functional annotation of 50 representative intact prophage genomes resulted in the identification of putative proteins such as bacteriocins, transporters, enzymes (hydrolases, catalases, phosphatases, lipases, to name a few), and others associated with immunity, sporulation, arsenate-, arsenite-, and bleomycin resistances as well as polysaccharide biosynthesis (Figure 6).

Some of these may affect *B. subtilis* in a desirable way from an industry point of view. For example, immunity-related proteins could be involved in defending *B. subtilis* from foreign phage infection [48]. Bacteriocin-related proteins could assist *B. subtilis* in having competitive advantages [14].Resistance-associated proteins, such as the arsenite resistance protein, may enhance host survival. A previous study involving the inactivation of arsenate and arsenite encoding genes in a skin element, a defective *B. subtilis* prophage, showed that they confer resistance to arsenate and arsenite [49].

On the other hand, prophages may also have undesirable effects. One such effect is the risk of interfering with the *B. subtilis*-mediated fermentation processes. Phage infections of starter cultures are serious risks in the food industry [50]. Moreover, agents that favor prophage inductions are common in food industries where *B. subtilis* may be used for various purposes. For example, UV-light, an established prophage inducer [16], is used for sterilization in food industries [51]. Likewise, some foods such as soy sauce, which are predominantly fermented by *B. subtilis* [52,53], are reported as being prophage inducers [54]. Desirable or not, it needs to be experimentally proven whether any of these putative genes play a role in the bacterial property and thus require further studies.

## Figures and Tables

**Figure 1 viruses-14-00483-f001:**
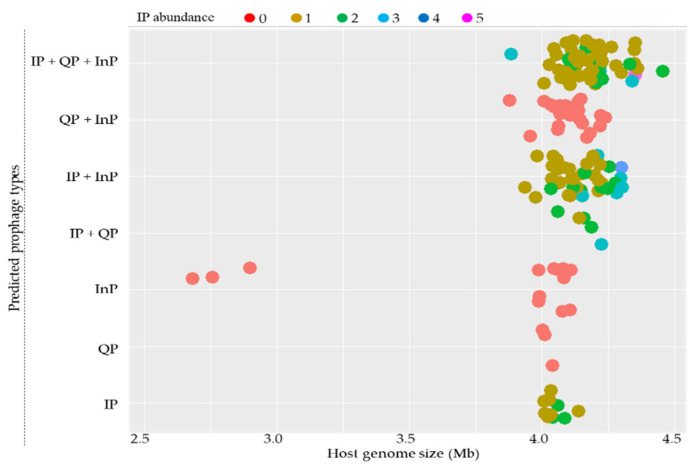
Distribution of intact prophages (IP), questionable prophages (QP), and incomplete prophages (InP) in 164 sequenced *B. subtilis* genomes. Host genome size is indicated on the *X*-axis. The distributions of the types of IPs according to host genome sizes are indicated on the *Y*-axis. Closed circles represent predicted prophages, and their colors indicate IP abundance, as indicated at the top.

**Figure 2 viruses-14-00483-f002:**
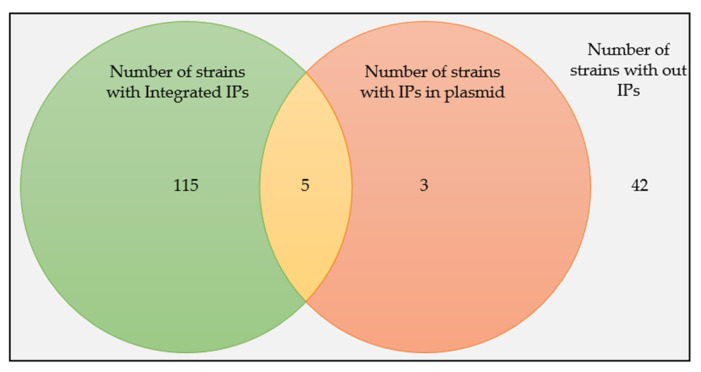
The prevalence of Ips in *B. subtilis* strains and their distributions as integrated or plasmid forms.

**Figure 3 viruses-14-00483-f003:**
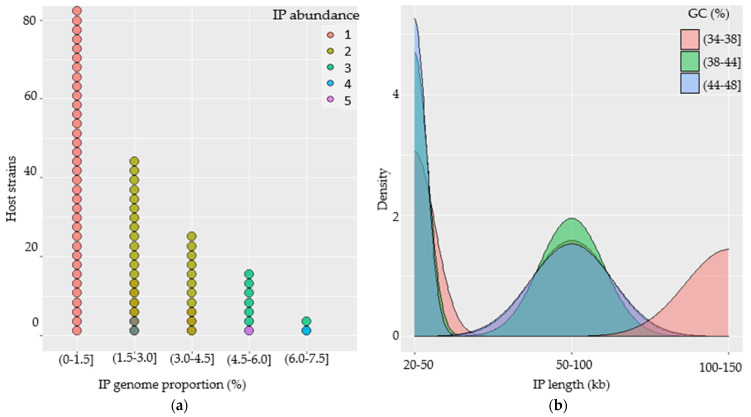
Characterization of intact prophages (IPs) in the genomes of *B. subtilis* strains. (**a**) Dotplot illustrating IP abundance and genome proportion per genome of their host and their associations. (**b**) Density plot showing the distribution and association of IPs in terms of their genome size and GC-content.

**Figure 4 viruses-14-00483-f004:**
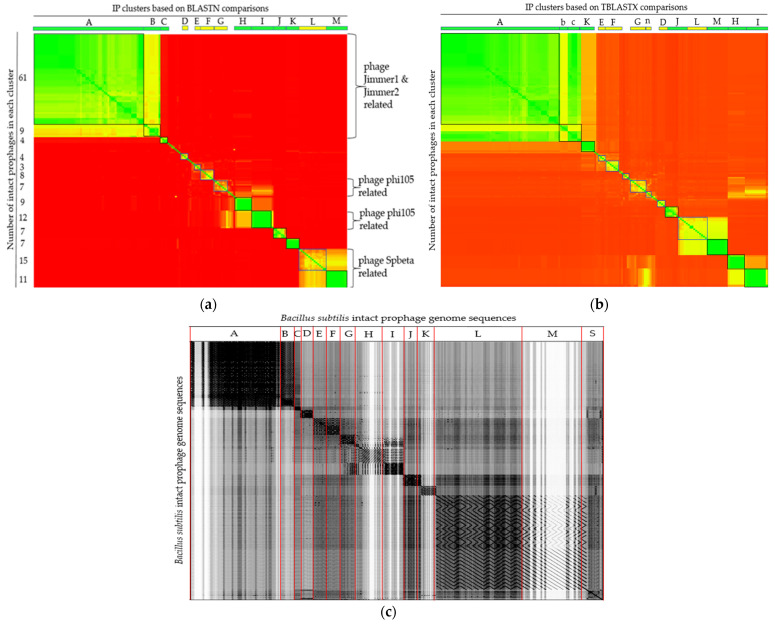
Comparison of 172 IP genome sequences predicted in *B. subtilis* (**a**,**b**) Gegenees heatplot generated using BLASTN and TBLASTX, respectively, with 200/100 settings. Heatplot colors indicate similarity ranging from nearly identical (green) to similar (yellow) or dissimilar (red). The corresponding numbers of group members of each cluster are indicated on the *y*-axis. Labeled boxes at the top represent IP clusters. (**c**) Dotplot alignment of IP sequences generated by Gepard.

**Figure 5 viruses-14-00483-f005:**
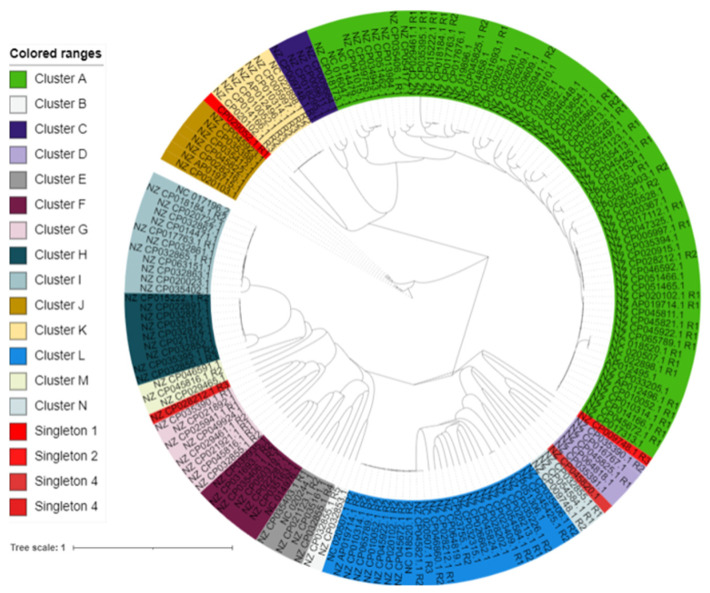
Phylogenomic relationships of *B. subtilis* IPs. The distances of the IP sequence were calculated using Gegenees. Then, the tree was constructed by SplitsTree4 and visualized by iTOL. Clusters are coded by different colors, and singletons are in red.

**Figure 6 viruses-14-00483-f006:**
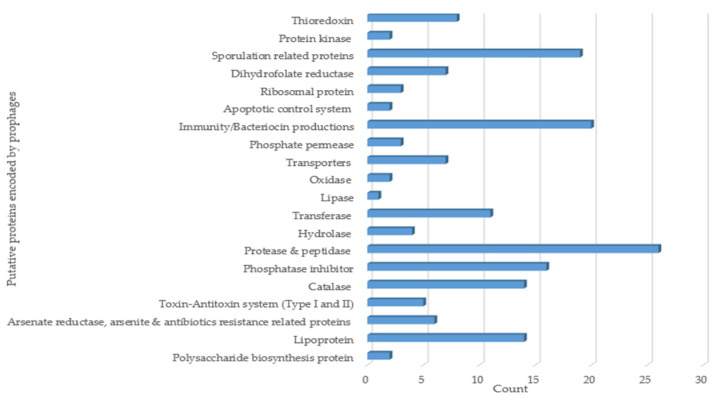
Functional annotation analysis of putative proteins encoded by *B. subtilis* prophages.

**Figure 7 viruses-14-00483-f007:**
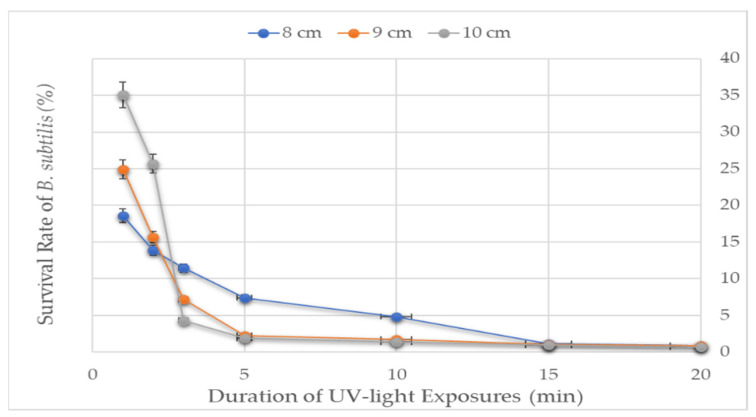
The survival rate of *B. subtilis* to UV-light treatment. UV-light tolerance of the *B. subtilis* was measured at three distances between the sample and the UV-lamp for seven varying exposure times.

**Figure 8 viruses-14-00483-f008:**
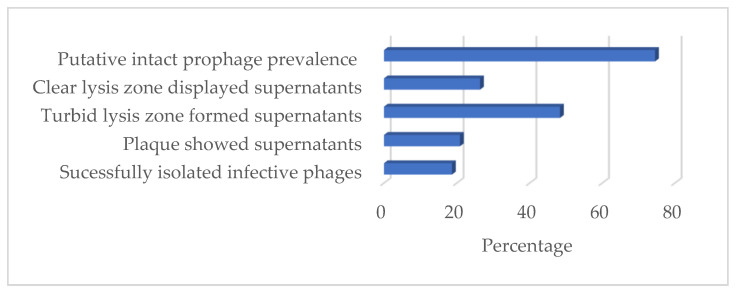
Prevalence of in silico-predicted prophages and UV-light inducible of prophages in *B. subtilis*.

**Table 1 viruses-14-00483-t001:** Genomic features of 171 intact prophages in *B. subtilis* predicted by PHASTER.

IP Genome Features	Mean ± SD	Min	Max
Length (kb)	55.02 ± 34.04	21.50	148.80
Genome proportion (%)	1.86 ± 1.54	0.01	7.18
Protein No.	71.33 ± 50.25	26.00	204.00
GC (%)	41.63 ± 3.90	34.32	47.75

**Table 2 viruses-14-00483-t002:** Genes present at 5′ and 3′ integrated intact prophage ends.

Clusters	Commonset 5′ and 3′ End Genes		Prophages’ Coordinates Exact Position with Respect to A Gene
	Most 5′ End Genes	Most 3′ End Genes	
A	ATPase YjoB	*N*-acetylmuramoyl-l-alanine amidase	5′ end → 3′ end
B	ATPase YjoB	*N*-acetylmuramoyl-l-alanine amidase	5′ end → 3′ end
C	Tetratricopeptide repeat protein	*N*-acetylmuramoyl-l-alanine amidase	Intragenic → 3′ end
I	Type I glutamate--ammonia ligase	Intergenic	5′ end → intergenic
F	Intergenic	Intragenic	Intergenic → intragenic
H	Intergenic	Intergenic	Intergenic → intergenic
M	SDR family NAD(P)-dependent oxidoreductase	Intergenic	5′ end → intergenic
K	Sigma-70 family RNA polymerase sigma factor	ImmA/IrrE family metallo-peptidase	Intragenic → intragenic

**Table 3 viruses-14-00483-t003:** The prevalence of UV-light inducible prophages in 5 *B. subtilis*.

Strain Name	Genome (Plasmid) Accession Number	Number of Predicted Prophages	No. of Lysis Showed Indicators Following Dot Assay		Plaque Display	Purified Phage
			Clear Lysis	Turbid Lysis		
NBRC 13719	NZ_AP019714.1 (NZ_AP019714.1)	2 (1)	5	2	Yes	Yes
NCIB 3610	NZ_CP020102.1	3 (1)	0	3	ND	No
SRCM102751	NZ_CP028217.1	1	2	9	No	No
ATCC 6633	NZ_CP034943.1	1	13	15	No	No
168	NZ_CP010052.1	3	0	3	ND	No

ND: Not detected.

**Table 4 viruses-14-00483-t004:** Phage host range analysis of *B. subtilis* phages on 28 *B. subtilis* indicator strains.

*B. subtilis* Indicator Strains	*B. subtilis* Phages (BSTP)																
	1	2	3	4	5	6	7	8	9	10	11	12	13	14	15	16	17
KCCM11779	+	+	+	-	+	−	−	−	−	−	+	+	+	−	−	−	+
KCCM35421	−	−	−	+	−	−	+	+	+	+	−	−	−	+	+	+	−
KCCM12248	−	−	−	+	−	+	+	+	+	+	−	−	−	+	+	+	−
KCCM41990	+	+	+	−	+	−	−	−	−	−	+	+	+	−	−	−	+
SRCM101407	+	+	+	−	+	−	−	−	−	−	+	+	+	−	−	−	+
SRCM100731	−	−	−	−	+	−	−	−	−	−	−	−	−	−	−	−	−
KCCM12513	+	+	+	−	+	−	−	−	−	−	+	+	+	−	−	−	+
KCCM11736	+	+	+	−	+	−	−	−	−	−	+	+	+	−	−	−	+
KCCM11734	+	+	+	−	+	−	−	−	−	−	+	+	−	−	−	−	−
KCCM11496	+	+	−	+	+	+	+	+	+	+	+	+	+	+	+	+	+
KCCM12513	+	+	+	−	+	−	−	−	−	−	+	−	+	−	−	−	+
KCCM40443	+	+	+	−	+	−	−	−	−	−	+	+	+	−	−	−	+
KCCM11733	−	−	−	−	+	−	−	−	−	−	−	−	−	−	−	−	−
KCCM41991	+	+	+	−	+	−	−	−	−	−	+	+	+	−	−	−	+
KCCM41462	−	−	−	−	+	−	−	−	−	−	−	−	−	−	−	−	+
KCCM11815	+	+	+	−	+	−	−	−	−	−	+	+	+	−	−	−	+
KACC 17802	+	+	+	−	+	−	−	−	−	−	+	+	+	−	−	−	+
KACC 12680	+	+	+	−	+	−	−	−	−	−	+	+	+	−	−	−	+
KCCM12511	+	+	+	−	+	−	−	−	−	−	+	+	+	−	−	−	+
70-4	−	−	−	−	−	−	−	−	−	−	+	+	+	−	−	−	+
KACC 17797	+	+	+	−	+	−	−	−	−	−	+	+	+	−	−	−	+
SRCM102751	+	−	−	−	+	−	−	−	−	−	+	−	−	−	−	−	−
KACC 10111	+	+	+	−	+	−	−	−	−	−	+	+	+	−	−	−	+
KACC 10112	−	−	−	+	−	+	+	+	+	+	−	−	−	+	+	+	−
SRCM100170	−	−	−	−	+	−	+	−	−	+	+	−	−	−	−	+	−
KCCM11796	+	+	+	−	+	−	−	−	−	−	+	+	+	−	−	−	+
SRCM100336	+	+	+	−	+	−	−	−	−	−	+	+	+	−	−	−	+
KCCM12027	+	+	+	−	+	−	−	−	−	−	+	+	+	−	−	−	+
Lysis (+)	19	18	17	4	23	3	5	4	4	5	21	18	18	4	4	5	19
No Lysis (−)	8	9	10	24	4	25	23	24	24	23	6	9	9	24	24	23	8

## Data Availability

The dataset that supports the central findings of this study are contained within the Appendix A.

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
