# Peer review of "Prevalence, Diversity and UV-Light Inducibility Potential of Prophages in Bacillus subtilis and Their Possible Roles in Host Properties"

_viruses, 2022, doi:10.3390/v14030483_

Round 1

Reviewer 1 Report

The present study of B. subtilis bacteriophages is the first which address all prophages of this bacterial species. It is mainly descriptive and holds potential for future detailed investigation of the identified prophages. The inducible and viable prophage candidates are of high interest to the scientific community.

The manuscript is well written, and the content is very accessible. Occasionally, I would like the authors to draw more conclusions from their data. However, since the diversity of phages studied is enormous, I understand why the authors proceed with such restraint.

I would recommend putting this study in the context of published findings. It should be mentioned that similar approaches have already been undertaken, focusing on SPbeta related phages in Bacillus genomes. Results revealed SPbeta to be widely distributed in the Bacillus genus (https://doi.org/s41396-020-00854-1). It should also be discussed, maybe as an outlook, that besides revealing the distribution of a phage-type and the prophage diversity, bioinformatical predictions can only serve as a starting point for subsequent investigations. In this sense, manual curation of the bioinformatical prediction of SPbeta revealed on the one hand that the automated prediction still includes many false positives and on the other hand that the identified SPbeta prophages hold more information as initially recognised. For example, this phage type's specific prophage integration sites became evident only after manual curation of the automatic prediction (https://doi.org/10.1101/2021.11.22.469490). Experimental verification of the identified prophages via UV induction adds further information. These procedures allow narrowing down the vast number of predicted prophages to the most interesting ones, which will serve as a starting point in future fundamental research.

Few minor comments:

Line 66: First attempts were already made to investigate the prophages of B. subtilis using publicly available sequence data, even focusing only on SPbeta. That should be mentioned in the introduction.

Line 193, 235, 299 and others… : B. subtilis should be in italic

Figure 1: It is not evident what the individual dots are. They should be more prominent and, if possible, also different in colour.

Line 308: Data from section 3.4 do not align with recent discoveries on SPbeta prophage integration sites. This fact should be addressed in the discussion.

Line 478: SPbeta

Reviewer 2 Report

The authors of the article report the effort of scanning for prophage sequences in the complete genomic sequence of B. subtilis deposited in public databases using two off-the-shelf prophage prediction applications. The work also reports an informal classification of all prophages found by the authors using sequence similarity and phylogeny methods. Finally, the authors induce phage generation via UV in strains of B. subtilis available by commercial and public biorepositories.

I believe a survey of phages observed in B. subtilis and a thorough characterization and classification of those phages as their viability if induced are interesting questions with significant interest to the Bacilus sp. community and phage specialist. The interest might be broader due to relevance of B. subtilis as a model organism and its use in the food industry.

But unfortunately, this work would need major revisions so I can confidently bless its publication in this journal.

My major concerns are the following:

The disconnect between the bioinformatics effort and the wet lab stage of the work. I would recommend the author to attempt to connect those two efforts, by, for example, performing the same bioinformatics analysis in the genomic sequence of the induced strains. Do growth rate and survival rate correlate with phage repertoire? Is there any correlation between inducible strains and the class of prophages present on those? Would the author consider collecting induced phages and sequence those? The discussion/conclusion section of the manuscript should describe the link between those two efforts (survey + wet lab).

Regarding the phage phylogenomic analysis, the author should indicate if his results agree or not with the current nomenclature/family of B. subtilis prophages. Are there divergences? Any special characteristic with the phages deemed singletons: size, number of genes, gene density.

A comparison (literature-based) of the author's findings with closely related pathogenic species of Bacillus would be very welcome too. Are there differences in the repertoire and number of phages?

Regarding minor flaws in the manuscript…

The author should perform a thorough review of the text. There are some typographic mistakes, verbs with incorrect verb tenses, and missing information to correctly interpret graphs and figures. A few examples:

  • Add X axis legend to fig 1.

  • Missing '. "Considering the position of both the 5’ and 3 end coordinates "

  • "**The** Gepard with a sliding window of 10 nucleotides was used to generate a dotplot alignment as shown in Figure 4c."

  • Table 2 - what “End” means in the third column.

  • Table 2 – “metallo- Begningopeptidase, depBegningent” Is this correct?

  • “In search for subtilis strains used in both the in silico and induction studies, we were able to identify 5 strains and ***analyzed***” (Table 3).

I also recommend the author to re-think some of the graphs in the article, it seems to me that those are unnecessarily convoluted. For example:

Figure 1 – Use color or shape to represent categorical data and axis to represent numerical data.

Figure 2 - Information can be represented by a venn diagram with only two circles, one representing IPs in plasmids, and the other representing integrated IPs. A rectangle surrounding both circles can be added. The complementary area of the rectangle would represent #strains without IPs.
